# Diet-Related Disparities and Childcare Food Environments for Vulnerable Children in South Korea: A Mixed-Methods Study

**DOI:** 10.3390/nu15081940

**Published:** 2023-04-18

**Authors:** Jiyoung Park, Seolhyang Baek, Gahui Hwang, Chongwon Park, Sein Hwang

**Affiliations:** 1College of Nursing, Institute for Health Science Research, Inje University, Busan 47392, Republic of Korea; pjy1113@inje.ac.kr; 2Department of Nursing, College of Nursing, WISE Campus, Dongguk University, Dongdaero 123, Gyeongju-si 38066, Republic of Korea; baekseolhyang@gmail.com; 3College of Nursing, Yonsei University, Seoul 03722, Republic of Korea; hwang_dawn@naver.com; 4Department of English Language and Literatures, Pukyong National University, Busan 46241, Republic of Korea; chongwonpark@pknu.ac.kr; 5Department of Social Welfare, College of Social Science, Inje University, Gimhae-si 50834, Republic of Korea

**Keywords:** child, diet-related disparities, food environment, mixed-methods study, socioeconomic disparities in health

## Abstract

Diet-related disparities that have often been observed in vulnerable families may play a negative role in children’s health and health-related quality of life. In South Korea, an afterschool care policy, called Community Childcare Center (CCC), was established in the 1960s to protect and educate vulnerable children; this role has expanded to provide meal services in recent times. Therefore, the CCCs’ food environment has become a pivotal platform for observing children’s nutrition and health-related disparities. Using a mixed-methods approach including a survey with self-reported questionnaires, field observation, and participant interviews, the food environment of CCC was explored alongside children’s eating behaviors. Eating behaviors were not as healthy as expected. Although service providers and cooks reported in the survey responses that the centers’ food environment was healthy, participant observations and interviews revealed a significant gap. Establishing a standardized food environment and improving the nutrition literacy of workers as a significant human resource at a CCC can promote healthy eating for vulnerable children. The findings suggest that in the absence of steps to improve the food environment of CCC, diet-related disparities may affect children’s health in the future.

## 1. Introduction

Diet-related disparities refer to differences in dietary intake, behaviors, and patterns in different segments of the population that result in poor dietary quality and inferior health outcomes for certain groups, and an unequal burden of disease incidence, morbidity, mortality, survival, and quality of life. The socioeconomic factors that cause children’s diet-related disparities include parental education level and family income; these factors are also powerful and influential contributors to health disparities [1,2]. The research on relationships between parental socioeconomic status and children’s eating behaviors and food consumption has consistently reported how these relationships affect health [3]. Differences in eating quality, stratified by household socioeconomic status, may cause inequalities in health status. These socioeconomic constraints are inevitable when providing nutrition to children [4]. Some researchers postulate that when such disadvantaged children become adults, they have an increased risk of obesity and metabolic diseases such as cardiovascular disease and type 2 diabetes [5,6].

Many studies have explored socioeconomic diet-related disparities in South Korea. Kim and Choo [7] observed the most vulnerable populations’ eating patterns and found a lower frequency of fruit and vegetable consumption than that of children in general populations, because children in poor households tend to find fast and low-cost food at home. Jang et al. [8] investigated elementary school students’ eating patterns and concluded that female students from low-income backgrounds preferred to frequently eat ramen (a type of popular instant noodles). Female students whose parents had low education and income levels consumed less plant-based food, fish products, vitamin C, calcium, and dietary fiber; both male and female students whose parents had a low education level consumed fewer fruits. Kim et al. [9] reported that infants and children from low-income families lacked micronutrition and tended to be overweight. An analysis of dietary safety found that approximately 10% of children aged 1–18 years in a vulnerable group consumed only one or two types of food, owing to a lack of food funds. In addition, 86.2% of households in this group skipped meals almost every month [10]. These children may experience parental absence related to poverty, and exposure to health-threatening behaviors, such as poor nutrition and a lack of exercise, resulting from unstable parenting and insufficient parental guidance on a healthy lifestyle [11]. As children from vulnerable families in Korea spend a considerable amount of time alone because of their parents’ economic activities, they often eat ramen and bread, skip breakfast, or fail to eat sufficiently, leading to binge-eating. Economic difficulties may also lead them to purchase relatively inexpensive instant food [12]. Therefore, urgent actions are required to improve eating behavior in this group of children.

Eating behavior is influenced by individual characteristics and diverse environmental factors in the places where people live and grow, including the home, school, workplace, and the community [13,14]. The various environments surrounding children can protect and prevent them from diet-related disparities during childhood [1]. However, among the various food environments of vulnerable children, inducing change in a family, including parental attitude and knowledge, is a challenging task because of socioeconomic restrictions. This may be related to parents’ low nutrition literacy, which is the ability to obtain, process, and understand nutrition-related information and the skills required to make appropriate nutrition decisions [15,16]. It may also be related to neglect or permissive parenting due to a lack of time and focus on earning money [12]. Therefore, when comprehensively considering the cultural and social background of children, it is necessary to select the preferred environment to inculcate dietary behaviors that are healthy and sustainable. Subsequently, we need to assess how this environment affects the child’s eating habits and what environmental changes should be induced to set and develop place-based initiatives or interventions.

In the 1960s, religious groups, individuals, and civic groups created small “study room” to protect and educate children in urban poverty-ridden areas in South Korea. These spaces spread rapidly with industrialization and urbanization. In addition to functioning as children’s study centers, the spaces included additional after-school care, catering, and counseling [17]. In 2004, the Korean government amended the Child Welfare Act to legislate Community Child Care Center (CCC) [18]. CCCs are childcare centers responsible for after-school care for socioeconomically disadvantaged children in South Korea. Currently, over 4000 CCCs nationwide are used by over 100,000 children under 18 years old from vulnerable families [19]. Among the services provided, meal services have an important physical, psychological, and social impact on children’s growth. The meal services help children form proper eating behaviors, provide psychological and emotional stability through meals, and promote physical health [20]. Thus, a CCC provides a significant environment for vulnerable children, and are known as a “second home” in South Korea [12].

The CCC is expected to create an environment that promotes children’s health and healthy living. However, because CCC started as a study center, most of them focus on teaching or caring for vulnerable children [21,22,23]. In addition, although South Korea has guidelines for kindergarten and school meal programs [18,24], there are no own standards for the CCC meal programs [25]. The Enforcement Decree of the Child Welfare Act in South Korea requires that nutritionists be deployed in CCCs with 50 or more children. However, because most centers comprise small groups of 20–30 children, only one CCC in Korea meets this criterion. All the other CCCs provide meals, including lunch box ordering, service providers directly cooking meals, and welfare program participants working as cooking assistants, based on the budget provided by local governments [18,26].

Although a few studies have attempted to identify children’s dietary behaviors and the food environment at CCC, most of them tended to use cross-sectional questionnaire survey tools [27,28,29] that make it difficult to identify the real nature of the CCCs’ food-related environment. Therefore, the purpose of this study was to investigate vulnerable children’s dietary behaviors and the food environment at CCC in South Korea using a convergent parallel mixed-methods design, which provides more breadth and depth of understanding of the research topic [30].

## 2. Materials and Methods

### 2.1. Study Design

We used a convergent parallel mixed-methods design [30] (Figure 1), wherein the quantitative and qualitative strands of the study were conducted independently and simultaneously, and combined the results in the overall interpretation. We administered a cross-sectional survey to CCCs’ children, service providers, and cooks (quantitative phase), and conducted participatory observation and interviews with participants at three CCCs (qualitative phase). We defined eating behaviors as children’s patterns of food and beverage consumption, as indicated by food choices, meal frequency, portion size, and so on, which we measured using a survey and observation. Food environments refer to the CCCs’ human-built and social environments, which influence children’s eating behaviors, measured through surveys, field observations, and interviews with participants.

### 2.2. Participants and Setting

#### 2.2.1. Quantitative Phase

We selected participants through convenience sampling of 17 CCCs in the community. Data were collected from 354 children, 34 service providers, and 15 workfare program participants who worked at the centers as cooks. The government assigns workfare program participants as center cooks at the center’s request. Different centers have differing roles for workfare program participants, but they commonly help with food preparation and serving and childcare. They perceive themselves as the “a mother figure of the center”, and thus provide a significant human influence that affects children’s eating behaviors at the center [26]. The data from five children who did not complete the survey were excluded; thus, 349 children, 34 service providers, and 15 cooks from 17 CCCs were included in the analysis. CCC eligibility criteria target children under 18 years of age who need priority care and who have been recognized by the local governor as needing care. Children in need of preferential care include those from vulnerable families, such as low-income households, households that include people with disabilities, multicultural households, and single-parent or grandparent households [31]. The participant characteristics of those included in the quantitative research phase are presented in Table 1.

#### 2.2.2. Qualitative Phase

Three of the 17 CCCs were included in the qualitative observations and interviews. These participants’ characteristics are also shown in Table 1.

### 2.3. Instrument

#### 2.3.1. Quantitative Phase

(1)Children’s eating behaviors: We used the Nutrition Quotient (NQ) [32] to investigate children’s eating behaviors. The NQ was developed by the Korean Nutrition Society, a leading organization in nutrition and health promotion in South Korea. Previous studies using large-scale samples have reported good reliability and validity for this instrument. The NQ has been used as a representative measurement for evaluating Korean children’s dietary behaviors [33,34,35]. It comprises 19 items divided into five categories: balance (five items), diversity (three items), moderation (five items), regularity (three items), and practice (three items). The balance factors include the intake frequency of cooked rice with whole grains, fruits, cow milk, legumes, and eggs. The diversity factors include the number of vegetables in each meal and the frequency of intake of kimchi and diverse side dishes. The moderation factors include the frequency of eating sweet foods, fast foods, ramen, late-night snacks, and street food. The regularity factors include eating breakfast, meal regularity, and time spent watching TV and playing computer games. The practice factors include chewing well, checking nutrition labeling, and washing hands before meals. Most of the evaluation items use a five-point Likert scale, but some use three- or four-point Likert scales. The scores were calculated by entering each answer into the Child Nutrition Index Program (http://www.kns.or.kr/, accessed on 3 February 2020), which calculates a nutrition index, grades, and scores for the five areas, where higher scores indicate better eating behaviors. Kim et al. [33] identified the following diagnostic cut-off points for the five NQ factors to detect poor nutritional intake using the receiver operator characteristic (ROC) curve analysis: balance (57), diversity (87), moderation (66), regularity (69), and practice (67). Scores below the cut-off points indicate poor nutritional intake. In our study, the NQ Cronbach’s alpha was 0.63.(2)Centers’ eating practices: Questionnaires were developed according to the Korean Ministry of Health and Welfare’s guidelines for healthy eating [36] (see Appendix A). The questionnaire measures how food is consumed and how healthy diets are pursued in the centers using 10 items scored on a five-point scale, where higher scores indicated that the center provides more nutritious food and promotes healthier diets. The Cronbach’s alpha for this scale was 0.771 in this study.(3)Service providers’ and cooks’ intentions regarding healthy eating practices: Self-reported questionnaires were used to measure the service providers’ and cooks’ awareness and risk perception, skills and self-efficacy, attitudes and outcome expectations, and social norms and support regarding healthy eating practices at the centers. This questionnaire was developed using the method suggested by Fishbein and Ajzen [37]. An example of an item related to skills and self-efficacy is, “I’m sure I can read the food labels,” and one related to social norms and support is, “People expect me to follow a healthy menu and recipe.” The questionnaire for the service providers and cooks comprised 22 and 24 items, respectively. Each item was scored on a seven-point Likert scale, ranging from 1 (disagree) to 7 (agree), where higher scores indicated better intentions of desirable behavior-related healthy eating at the centers. In our study, the Cronbach’s alpha for service providers and cooks was 0.875 and 0.852, respectively.

#### 2.3.2. Qualitative Phase

The participant observation and interview items are shown in Appendix B. We observed meal and snack preparation and consumption, which included food preparation, cooking, distribution, and eating.

### 2.4. Data Collection

The Institutional Review Board of the researchers’ university approved this study (2019-01-011-001).

#### 2.4.1. Quantitative Phase

Data were collected from 18–28 February 2019. With the help of the CCCs’ community council, the researcher telephonically contacted 17 CCCs to explain the study purpose, and participation was confirmed by all 17 centers. Children’s individual data were collected only for children whose parents voluntarily agreed to their children’s participation. We established a data collection protocol to maintain consistency, which detailed the data collection materials, survey methods, and precautions. One service provider from each center was selected as a data collector; we provided them with data collection training based on the data collection protocol. Completing the survey took approximately 10 min for children and 5 min for service providers and cooks.

#### 2.4.2. Qualitative Phase

We chose to include a qualitative approach because it facilitates better field research. With a context-sensitive approach, qualitative researchers can thoroughly explore the chosen topics [38]. We used participant observations and interviews to examine the children’s eating behaviors and the CCCs’ eating environments. The observation and interview details are provided in Appendix B. We chose three CCCs for the qualitative research who willingly wanted to participate in the participant observation and interviews. Prior to data collection, the participants were informed of the study purpose, intention, and research process, and assured of data anonymity and confidentiality; the participants provided written informed consent. Following the participant observation, we provided gift certificates to the center (worth about 50 US dollars) as a token of appreciation.

### 2.5. Data Analysis

#### 2.5.1. Quantitative Phase

We used IBM SPSS Statistics, version 21.0 for the data analyses. All the data were screened to confirm their accuracy and ensure that the statistical tests’ assumptions were met. We conducted descriptive statistics, independent t-tests, and analysis of variance (ANOVA) to analyze the quantitative data.

#### 2.5.2. Qualitative Phase

We used NVivo 12 to conduct a content analysis of the transcribed data. Four researchers independently analyzed the qualitative data. They coded words, phrases, or paragraphs relevant to the research questions and categorized the initial coding according to their meanings. Themes and subthemes emerged from the raw data as the analysis progressed. The qualitative research analysis was conducted according to the Consolidated Criteria for Reporting Qualitative Research [39].

## 3. Results

### 3.1. Quantitative Findings from the Survey

#### 3.1.1. Diet Quality and Eating Behaviors

The Korean version of the NQ showed that the average total score was 58.95 out of 100 points, whereas those of the questionnaire’s five sub-factors were 67.91 for diversity, 67.21 for moderation, 58.99 for regularity, 57.33 for practice, and 50.26 for balance. Considering the cut-off point for each sub-factor, “moderation” was the only sub-factor to exceed the cut-off point (see Figure 2). When a further analysis was conducted, children who attended the CCC “often” (more than 5 days/week) and those who perceived their physical condition as “very healthy” showed significantly higher scores on the NQ compared with their counterparts (see Appendix C).

#### 3.1.2. CCC Workers’ Perceptions of their Food Environments

Both service providers and cooks reported that the center’s eating environment or practices were healthy (see Table 2). The service providers reported higher scores for items 1, 6, 7, 9, and 10, and the cooks scored 4.93 points out of 5, especially for items regarding distributing the appropriate amount of food (item 8) and working together to provide a healthy meal (item 10). Although there is no statistical comparison of the scores between the service providers and cooks, it is likely that the cooks scored higher for all items (Table 2).

As shown in Table 3, both the service providers and cooks reported mostly positive intentions regarding their CCC’s eating practices. However, the service providers showed a relatively low awareness of menu and recipe use, identifying healthy food ingredients, rejecting children’s requests for extra meals, and eating together (items 9, 11, 13, 14, 17, 18, and 20). Similarly, the cooks reported a relatively low intention to monitor and reject children’s requests for extra meals and eat together (items 2, 3, 17, 18, and 20).

### 3.2. Qualitative Findings from the Field Observation and Participant Interviews

The field observations and interviews with the service providers and cooks revealed several common characteristics that could increase diet-related disparities. First, the children were observed eating unhealthy foods freely, but the healthy food options for them were very limited. Second, not all CCC workers had sufficient nutrition literacy to properly feed vulnerable children. In addition to these food environment challenges, the external social support was not conducive to decreasing disparities because unhealthy foods were donated. In addition, undesirable management styles, such as a permissive or authoritarian atmosphere, were observed, along with poor parenting styles, prompting children’s unhealthy eating.

#### 3.2.1. Free Access to Unhealthy Foods but Limited Access to a Healthy Diet

All the CCCs allowed children to snack at will between meals and appeared to have no set plan for this; therefore, children were affected by the type of food available to them daily. No consideration was given to foods’ calorie content; therefore, high-calorie foods containing trans fats, such as ready-made hot dogs, packaged chocolate pie, creamy breads, or donuts, were often given as snacks or dessert. The center directors predominantly controlled the purchasing of food ingredients. All the CCCs tended to buy food ingredients that were not fresh and instead bought cheap and pre-processed bulk food packages.

One-way passive communication, such as the cook writing a memo below the menu board, was typically used when there was a need to communicate regarding food ingredients. In addition, the workers tended to change the daily menus according to the food availability in the kitchen and the centers’ contextual constraints. Fresh fruits and vegetables were rarely seen at the CCCs, unless they were externally donated. In some cases, children were given high-calorie snacks as an incentive to study hard.


*I studied science very hard today, so I got a snack from the center in return for my hard work.*
(*Obese girl; Center A, Participant observation statement*)


*What this center director usually buys is the cheap-for-quantity or bulky quantities. No matter the expiration date, there is lots of frozen meat in the fridge for a very long time.*
(*A cook; Center B, Participant observation statement*)

#### 3.2.2. CCC Workers’ Poor Nutrition Literacy

The centers normally displayed the food menus on their kitchen wall alongside a sheet listing the type and status of food ingredients in the kitchen. Monthly menus for daily eating were officially set by the local government as part of the Nutritional Education Program for Children and the CCCs were obligated to follow them. The cooks tended not to follow recommended cooking methods, changing the methods to encourage the children to eat more. One cook reported that when they did not know how to cook an item, they usually asked the director or other staff members; otherwise, they gave up on following the menu. They repeatedly said that the children were extremely reluctant to try food they had never eaten before; therefore, they had difficulties with the children when the menu contained new ingredients. Although the centers had several types of measuring cups and gauges to measure ingredient quantities, they said that using that kind of tool was not their job but was for young professionals at a cooking school. They also said that trying to use the measuring cups was uncomfortable.


*Yesterday, in our menu, we had Spergularia marina Griseb (sebal herbs), which I have never heard of… It was tough for me to purchase this in the market, and the children did not eat it… The center director and I agreed that we should substitute the herb with spinach in our menu. The same is true for curled mallows included in the menu… Radish is good for steamed dishes because it makes you feel refreshed… This is the way I cook. People rarely use radish when cooking steamed dishes…*
(*Center B, Individual interview with a cook*)


*I never try to use measuring cups, but this sort of thing must be taught in cooking schools… I am too old to learn it… If they fire me, I will have no place to work… Younger people attend cooking schools…*
(*Center B, Individual interview with a cook*)

#### 3.2.3. Permissive or Indulgent Atmosphere: Only Seen at Centers A and B

After the meal preparation was completed, the service providers had the children line up in front of the food distribution tables, in a given order, after washing their hands. In Centers A and B, the first meal reflected what the children requested, and they were allowed to eat as much as they liked. During the meal distribution, both the director and social worker continued working in their office. The atmosphere in the dining area was generally distracting. Some children compared their food amounts with that of their friends or complained of inconveniences caused by overeating. During this time, the service providers returned to their workplace, or sat and ate their meals separately from the children. The cooks washed dishes, cleaned up, or ate in the kitchen. The workers seemed to believe that their job requirement was only providing food.


*Three seemingly fat children (one girl and two boys) gathered for a meal. More than two distributions had already been made. One of the boys licked the plate with his tongue to eat the soup. A child’s sister finished one helping and returned to her seat with as much rice and soup as was taken the last time. It appeared to be a lot, especially as she only took rice. The child was observed crawling in the kitchen and said, “I can’t move because I’m so full.” Further, the girl took five helpings of food.*
(*Center A, Participant observation statement*)


*A teacher came to the room to eat and said, “Sir, time to eat.” The service provider replied, “Go ahead. I am in the middle of taking care of paperwork.”*
(*Center A, Participant observation statement*)


*The child did not have any vegetables on his plate. The service provider said, “Why don’t you try some vegetables?” The child pretended not to hear this. When she tried to give him soup with vegetables, the boy said, “Nope. No vegetables!” The worker hesitated, but she asked, “Why not the vegetables?” The child only took the soup. When the service provider gave some vegetables with soup to another girl, she said, “Don’t put the vegetables on my plate!”*
(*Center B, Participant observation statement*)

#### 3.2.4. Authoritarian Atmosphere: Only Seen at Center C

A service provider said that she felt it was her duty to feed the children vegetables as much as possible, because some parents did not cook vegetables, but usually provided meat at home. However, even if the service providers knew how to properly feed children vegetables, such as by mincing or roasting with other ingredients, they simply tended to force children to consume all the vegetables given. Owing to these forceful attempts, it was observed that a child had to keep chewing and had difficulty swallowing the food. Another child was observed standing up after eating only rice, while keeping his head down to look at his plate with the remaining vegetables.


*If you let children eat what they like, there’s a lot they don’t eat. I need to fix that… We try to make them eat even if they are about to vomit. It is not easy to do so if they are picky eaters! Especially for the lower grades; they are picky, and we try to fix their bad eating behavior all the time.*
(*Service provider; Center C, Participant observation statement*)

#### 3.2.5. Inappropriate Social Support: Unhealthy Donated Foods

The food menus were often affected by the food donated to the center. The food bank, for example, usually donates sweetened breads, carbonated soft drinks, and fish cakes with lots of salt and oil. Some companies donate large amounts of food, including kimchi, which is subsequently stored in the kitchen for at least several months. Whenever food comes without advance notice, it is used for impromptu main meals or snacks. When too much food is donated, the centers redistribute it to the children’s homes.


*“Today, fish cake was donated to us. You can take it when you go home. One plastic bag per person! Do not leave it in your bag, otherwise it goes to your school tomorrow. As soon as you go home, please give it to your mother. Don’t forget to take the boiled fish paste with you, guys!” said workers*
(*Center A, Participant observation statement*)

#### 3.2.6. Poor Parenting Style at Home Prompts Unhealthy Eating

The field observation trials did not provide an opportunity to visit the children’s homes; nonetheless, a few workers reported that most of the children spent time alone at home. Some of the children were often seen with small amounts of pocket money given by their parents, which led them to buy snacks or instant noodles as often as they wanted at convenience stores. In addition, the children were often observed overeating or eating quickly at the center, owing to starvation and inadequate food intake at home. These phenomena are related to a lack of parental care.


*The service provider said, “The children ate snacks before entering the center… Parents gave their children two thousand (Korean) won every day to assuage their regret for not taking good care of them… Then, the children ate snacks with the money… Some did not eat anything at the center and ate cup ramen at home. I warned parents not to give money to their children. They said that they cannot help it because they always feel sorry for them…”*
(*Center B, Participant observation statement*)


*After the children were served several times, the child said, “Teacher, my sister is porky, and I am piggy. I have not eaten all day. I only ate a little bread at home.”*
(*Center A, Participant observation statement*)

## 4. Discussion

In this study, the NQ score of children at CCCs was 58.95 out of 100 points, which is lower than those in a study of Korean (64.99) and Chinese (66.57) elementary school students [40] and in a study of elementary school students in Korea (boys: 61.40, girls: 63.02) [41]. As both studies [40,41] were conducted in elementary schools with children belonging to the general population, it can be inferred that children at CCCs have unhealthier eating behaviors than the general population. Children’s negative eating behaviors were vividly observed in our qualitative results as well. As emphasized in the introduction, from the perspective of diet-related disparities [1,2], systematic efforts to establish healthy eating behaviors from an early age for vulnerable children should be prioritized and sustained.

Further, in this study, only the moderation score among the five sub-factors of NQ exceeded the cut-off point. Drawing from the sub-group analysis, the younger the age and the more frequent the visits to the CCC, the higher the moderation score. In previous studies, it was thought that parents have more control over their children’s food intake during early childhood than during elementary school, as children in elementary school can freely choose food items [42]. Furthermore, it is possible to infer the organizational effect of CCCs on children’s moderation. Many studies have reported that after-school care organizations or programs such as the CCCs in South Korea have a positive protective effect on children’s health and corresponding behavior [12,43,44,45]. However, the research designs and collected data structure limit our findings regarding the protective organizational effects on children’s eating behaviors in consideration of various exogenous variables. Future studies should investigate organizational effects on children’s eating using hierarchical linear modeling with longitudinal and nested data. Additionally, it would be interesting to investigate how the CCCs’ characteristics, such as the type of food distribution and the center director’s attitude, affect children’s health behaviors and health.

Both service providers and cooks indicated in their survey responses that, “We can read and follow the menu and recipe correctly.” However, the participant observation showed that the menu was either changed or the food was cooked without referring to a recipe. This may be because local governments only offer menus every month and do not provide instructions or manuals for recipes; therefore, workers often do not know how to properly prepare the food. In Japan, some local governments, such as the Chiba Prefectural Government, provide recommended meals and recipes on their websites that follow seasons and targets; moreover, children’s facilities write recipes and provide cooking tips, such as how to cut ingredients, cooking order and seasoning, and heating appliance time settings based on the guidelines for “Dining in Children’s Welfare Facilities” [46]. These menus and recipe-based meals can be evaluated by the children to ensure that nutrition and flavor are maintained [47]. Therefore, cooking-related education needs to be strengthened so that standardized meal manuals can be prepared and followed. In addition, a meal service evaluation system should be established, such as conducting regular menu satisfaction surveys for children. Considering the vulnerability of CCC cooks (20% with an elementary and 6.7% with a middle school education, in our sample), cooks’ low nutrition literacy level should be accounted for when developing detailed food preparation guidelines for them.

The center workers’ survey responses indicated that they ate together and were aware and convinced of the importance of dining together. They also indicated that they recognized the importance of monitoring children’s eating behaviors and environment. However, this perspective was not implemented in reality according to the qualitative results. The primary human environments that surround children, such as childcare providers, need to teach children “proper” table manners for healthy eating [48]. Service providers’ modeling of desirable eating behaviors, active guidance, or education regarding healthy foods, and restrictive guidance or enforced rules regarding food intake can increase children’s intake of healthy food and reduce that of unhealthy food [49,50,51]. That is, the eating behaviors of childcare providers as primary role models can affect children’s eating, and there is a need to educate them. For example, Head Start, the representative policy to provide a comprehensive program to low-income children and families in the US, created a “Head Start Performance Standards” program to help service providers learn healthy eating behaviors [52]; in addition, the American Dietetic Association suggested an educator’s guide for children’s healthy eating [53].

Permissive or indulgent environments greatly influenced the amount of food children consumed at the CCCs. Both the service providers and cooks indicated in the survey that their centers distributed adequate amounts of food to the children and refused to provide more, and that the children refrained from overeating and consumed appropriate amounts. However, the participant observations revealed that children often ate high-calorie snacks and took three or four helpings, focusing on their preferred side dishes. Moreover, they sometimes complained of discomfort caused by overeating, especially in Centers A and B. As childhood obesity is closely related to food portion size [54,55], service providers need to ensure that the proper amount of food is distributed during each meal to promote nutritional balance through the even ingestion of various foods. CCCs’ unhealthy food environment may be driven by complex factors, such as an overload of administrative work, aid worker scarcity, unfriendly meal guidance, a flawed nutrition management system, and workers’ low nutrition literacy. Thus, there is an urgent need for developing and distributing a standardized system and manual to create healthy food environments at such centers. The system focus should shift from records that are dependent on the service provider to user-friendly records for the easy management of individual children’s meal consumption. In addition, education programs are required to enhance CCC workers’ ability to obtain, process, and understand nutrition information, and develop the skills to make proper nutritional decisions.

An authoritarian atmosphere may also be a negative environment for children’s healthy eating. The service providers’ authoritarian style at Center C was unlike that of Centers A and B. Many studies have reported the negative impact of excessive control and pressure on children’s diets [49,56,57]. For example, attempts by parents to forcefully feed their children cause stress for both the parents and their children, which negatively affects the child–parental interaction [56]. It also causes side effects that alter children’s food preferences [49]. While it is challenging for childcare settings to select and distribute food to large numbers of children while considering individual preferences, it is important to guide children to healthy alternatives, rather than forcing or restricting their diets at childcare centers. It is important to understand and communicate with the child to understand their consumption behaviors rather than forcing an adult’s perspective of nutrition on them [58]. Once the patterns are comprehended, they should be conceptualized so that children can feel independent while choosing what they eat. Strategies, such as mixing new foods or food that children generally dislike with foods they like, should be implemented to expand children’s palettes and help them enjoy a wider variety of foods [59].

Childcare food environments are also affected by other environmental factors, such as the child’s home, food donations, and the community. That is, the accessibility, availability, and adequacy of food within a center can be affected by factors outside the institution [60,61]. Thus, to induce children’s healthy eating behaviors, intervention in the CCCs’ external environments is required alongside changes within the organization.

The first channel to consider in socioeconomically vulnerable children’s childcare facilities is food donation. We observed that excessive numbers of donated fish cakes were eventually delivered to families through the children. Furthermore, in this study the item scores related to donation were lower than the other survey item responses for workers. This may indicate that they might have difficulty choosing which food is donated. This is related to the food bank’s initial activities. The government’s policies and public relations are heavily oriented toward food waste reduction; users’ needs and circumstances are not fully considered [62]. Additionally, the types of donated foods are limited; most of them are processed or fast foods rather than fresh ingredients, which does not guarantee the food’s appropriateness [63]. In USA, the Institute of Medicine of the National Academies developed and published nutrition guidelines for charitable food systems, titled “Nutrition-Focused Food Banking”; the health food list they developed was actively promoted with charities to improve the quality of food chosen by food bank users [64]. Donations are triggered by the perception that CCCs are places for children from a low-income background; however, they should be provided in response to user needs, not for benefits or just charity. Thus, it is necessary to prepare and promote guidelines for improving donors’ awareness and playing an active role in the public and private sectors.

This study shows that children’s eating behaviors and childcare food environments are also affected by the home environment. Therefore, it is necessary to consider the eating environment at home, especially with family members. That is, family education on healthy eating should be strengthened, and a manual for handling these issues should be published to establish order. Unfortunately, these problems cannot be completely solved simply by family education or emphasizing responsibility. That is, the fact that caregivers are unable to share meals or monitor their children because they need to earn a living suggests that a structural factor underlies the issues related to adequate care for children [12]. Furthermore, enhancing healthy foods’ accessibility and availability to children in the community has a major impact on children’s diet and health [65]. Therefore, the kinds of food children eat outside of the center and what foods are limited is closely related to healthy eating behaviors. In this way, as the environments inside and outside the center are interconnected, the CCCs can serve as a significant platform for various environmental stakeholders surrounding vulnerable children to become healthy together and ultimately prevent diet-related disparities in South Korea.

Although this study was conducted before the COVID-19 pandemic, it suggests an urgent need for the active development of policies and programs for children from vulnerable families who are exposed to the socioeconomic diet-related disparities that are intensifying during the COVID-19 era. Children from vulnerable families who are highly dependent on school meals are more prone to these disparities. Considering these issues, during the COVID-19 pandemic, the US government provided meals through the National School Lunch Program and the School Breakfast Program [66]. Additionally, “telehealth,” including tele-exercise and tele-nutrition, provided innovative and effective opportunities to promote healthy lifestyles for vulnerable children during the COVID-19 pandemic [67]. The effectiveness of these policies and programs suggest that it is necessary to increase healthy food availability and accessibility for vulnerable children during the recurrent pandemic, and to activate non-face-to-face online education for vulnerable children, families, and after-school program caregivers to improve nutrition intake and alleviate socioeconomic diet-related disparities.

## 5. Conclusions

This study attempted to explore children’s eating behaviors and food environments in childcare centers for socioeconomically vulnerable children. The findings showed that to ameliorate inadequate diet-related practices and promote healthy living, the development and distribution of manuals that specify and standardize matters related to cooking, meal distribution, and management, including staff training, and the improvement of the donation culture of CCCs were essential. The mixed-methods approach, which incorporated quantitative and qualitative methods, enabled the researchers to identify the lacuna between how service providers perceived reality and what was implemented in their daily routines. Children’s eating behaviors are affected by various environments, such as those at home, school, and CCCs; these can affect the diet-related disparities among children in South Korea. While the family is a vulnerable and difficult environment to change among children’s environments, the school has already provided a standardized food environment that is optimal for all children. Therefore, the CCCs provide a unique environment targeting only vulnerable children in South Korea and are the optimal environments to set and provide place-based initiatives or intervention for vulnerable children. In this study, the organizational food environment of the centers was found to be unhealthy. Considering the changeability of the environment and the potential impacts of interventions, creating healthy food environments at CCCs is the most significant step to eliminate diet-related disparities and ultimately improve the health of children. The lack of reliability of the NQ may be a limitation of this study. However, as a representative tool for measuring children’s eating behaviors in South Korea, the reliability of this tool has already been confirmed [32]. In addition, the use of the NQ enables comparisons with other children’s groups. This study did not directly address the perspectives of other stakeholders, such as parents, school teachers, and policy makers, who may impact children’s eating behaviors and food environments. Therefore, further studies that examine other stakeholders’ roles are necessary.

## Figures and Tables

**Figure 1 nutrients-15-01940-f001:**
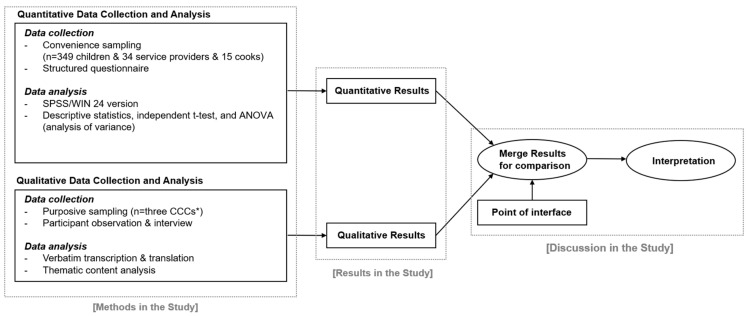
Research approach: Convergent parallel design [30]. ***** CCCs: Community Childcare Centers.

**Figure 2 nutrients-15-01940-f002:**
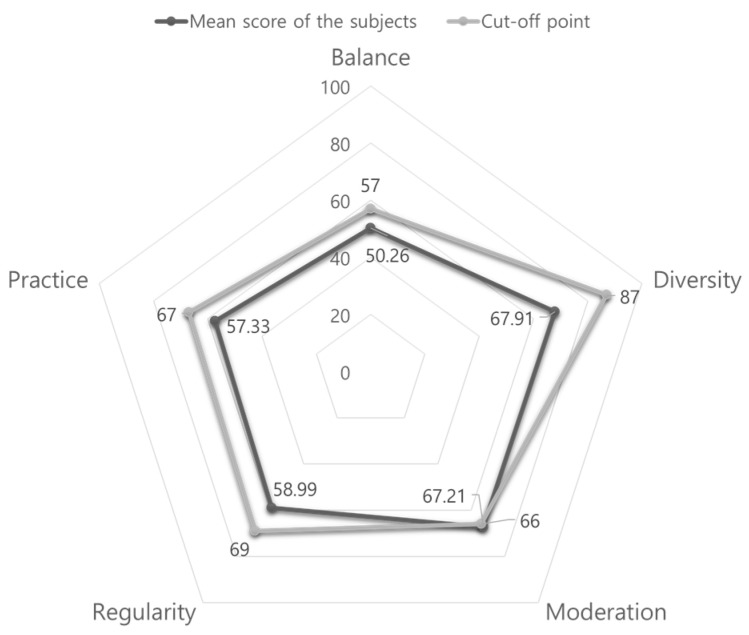
NQ sub-factor scores and their cut-off points (*n* = 349).

**Table 1 nutrients-15-01940-t001:** General characteristics of the children and workers who enrolled in the mixed-method study.

Type of Research	Subjects	Variables		N (%) or Mean (±SD)
Quantitative study	Children (*N* = 349)	Age (year)		10.2 (±2.4)
		Sex		
		Boy		175 (50.1)
		Girl		174 (49.9)
		Socioeconomic status		
		Basic living security ^1^		101 (28.9)
		Second-lowest income bracket ^2^		43 (12.3)
		Childcare exception ^3^		94 (26.9)
		Others ^4^		111 (31.7)
		Nutritional status		
		Underweight		11 (3.2)
		Normal		220 (63.0)
		Overweight		47 (13.5)
		Obese		71 (20.3)
		Duration of CCC attendance		
		6 months–less than 1 year		102 (30.5)
		1 year–less than 3 years		103 (36.8)
		Longer than 3 years		109 (32.6)
		Weekly frequency of CCC attendance		
		3 or 4 days		31 (9.3)
		5–7 days		304 (90.7)
		Perceived body image		
		Skinny		103 (30.8)
		Normal		144 (43.0)
		Obese		88 (26.3)
		Perceived physical condition		
		Very healthy		104 (31.0)
		Healthy		196 (58.5)
		Unhealthy		33 (9.9)
		Very unhealthy		2 (0.6)
	Service providers (*N* = 34)	Age (year)		45.5 (±11.2)
		Sex		
		Male		4 (11.8)
		Female		30 (88.2)
		Education		
		College		26 (76.5)
		Graduate school		7 (20.6)
		Other		1 (2.9)
		Length of service at current CCC (month)		52.8 (±40.4)
	Cooks (*N* = 15)	Age (year)		53.9 (±6.1)
		Sex		
		Female		15 (100)
		Education		
		Elementary school		3 (20)
		Middle school		1 (6.7)
		High school		11 (73.3)
		Period of working at current CCC (month)		22.8 (±20.4)
Qualitative study	CCCs ** (*N* = 3)	Number of children	Center A	24
			Center B	29
			Center C	31
		Number of workers	Center A	3
			Center B	3
			Center C	4
		Current CCC operation period (year)	Center A	15
			Center B	14
			Center C	9

^1^ A child whose parents are unable to provide for basic family support, including livelihood, housing, and education, according to the National Basic Living Security Act (Article 14-2); ^2^ A child who is not eligible under Article 14-2 and whose family’s countable income is below the criteria prescribed by Presidential Decree; ^3^ A child who is recognized by the head of a district as requiring care because of difficulty at home; ^4^ A child whose parent is an immigrant worker with a language barrier, or a single parent, grandparent, and so on [31]. ** CCCs: Community Childcare Centers.

**Table 2 nutrients-15-01940-t002:** CCCs’ eating practices reported by service providers and cooks.

Item	Service Providers (*n* = 34)		Cooks (*n* = 15)	
	Mean **	*SD*	Mean **	*SD*
1. Our center promotes a balanced diet.	4.53	0.507	4.86 *	0.363
2. Our center discourages overeating.	4.15	0.857	4.60	0.507
3. Our center rarely provides unhealthy foods (i.e., salty, sweet, or greasy food.)	4.44	0.660	4.80	0.414
4. Our center provides plenty of drinking water.	4.12	0.769	4.60	0.737
5. Our center prepares only as much food as planned.	4.35	0.646	4.73	0.594
6. Our center cooks food according to the menu.	4.59	0.557	4.73	0.458
7. Our center uses fresh food ingredients.	4.50	0.663	4.80	0.414
8. Our center distributes the proper number of meals for children.	4.35	0.485	4.93	0.258
9. At our center everyone dines together.	4.59	0.783	4.60	1.056
10. Staff at our center help each other to provide healthy food for children.	4.50	0.564	4.93	0.704

* One missing case was included (*n* = 14). ** 5-point Likert scale: 1 strongly disagree, 2 disagree, 3 neutral, 4 agree, 5 strongly agree.

**Table 3 nutrients-15-01940-t003:** Service providers’ and cooks’ intentions toward eating practices at CCCs.

Item	Service Providers (*n* = 34)		Cooks (*n* = 15)	
	Mean *	*SD*	Mean *	*SD*
1. Monitoring children’s eating habits is important.	6.41	0.701	6.00	1.464
2. Monitoring the food environment of the center is necessary.	6.00	0.921	5.67	1.496
3. Monitoring of the snacks and food eaten by children at the center is needed.	6.03	0.870	5.93	1.223
4. I must make sure that the children eat healthily at the center.	6.68	0.589	6.80	0.414
5. For obesity prevention and health promotion, healthy eating must be ensured by the center.	6.26	1.024	6.73	0.458
6. People expect me to make sure that children eat healthily at the center.	6.26	0.828	6.53	0.834
7. For healthy eating, children need support from a trusted institution.	6.44	0.824	6.87	0.352
8. I understand the need for a healthy meal.	6.65	0.597	6.67	0.488
9. I can read the menu and the recipe correctly.	5.94	0.886	6.53	0.743
10. People expect me to follow a healthy menu.	6.15	0.784	6.67	0.488
11. I can comply with a healthy menu and recipe.	5.91	0.965	6.53	0.743
12. I can purchase the proper amount of food ingredients for the menu.	6.06	0.814	-	-
13. I can separate harmful foods from foods donated to the center.	5.85	1.048	-	-
14. I can read the food labels.	5.68	1.007	6.53	0.640
15. I can discuss food taste with the center’s members on a regular basis.	6.18	0.834	-	-
16. I can feed the center’s children with the proper amount of food.	6.09	0.933	6.87	0.352
17. I know how to say no to a child who wants a second serving of food at the center.	5.24	1.597	5.67	1.234
18. I know that all members of the center can dine together.	5.97	1.314	5.80	1.781
19. For healthy eating, children need to eat together.	6.47	0.825	6.53	0.834
20. People expect me to eat with children at the center.	5.88	1.008	5.87	1.727
21. Children need to follow healthy dietary behaviors at the center.	6.59	0.609	6.87	0.352
22. I am sure that children eat healthy foods at the center.	6.18	0.834	6.73	0.594
23. I can distinguish healthy food ingredients for cooking.	-	-	6.80	0.414
24. I do not cook unhealthy foods (i.e., salty, spicy, sweet ones).	-	-	6.67	0.488
25. I can make healthy and tasty food(s) for children at the center.	-	-	6.53	0.834
26. I can discuss cooking with the service providers.	-	-	6.60	0.910
27. I do not follow unhealthy cooking methods.	-	-	6.87	0.352

* 7-point Likert scale ranging from 1 (disagree) to 7 (agree).

## Data Availability

The data presented in this study are available on request from the corresponding author. The data are not publicly available due to the ownership of the data.

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
