# Peer review of "Diet-Related Disparities and Childcare Food Environments for Vulnerable Children in South Korea: A Mixed-Methods Study"

_nutrients, 2023, doi:10.3390/nu15081940_

Round 1

Reviewer 1 Report

the introduction, discussion and conclusions are too long paragraphs.  
the authors must try to synthesize the concepts.  reading can become boring and incomprehensible.  too much information.  try to shorten periods that are too long.

Results.there are too many tables in the results, the authors should include some results in the supplementary material.  for example the photographs.  It's not good quality.

the English language is unscientific, the authors should have the whole manuscript revised in a less scholastic English form.

conclusion. The autore should emphatyze the most important results in less sentences.

Reviewer 2 Report

The work very well describes the context of nutrition organization and legal regulations in Korea, both quantitative and qualitative data were used. A few notes below:

  if exclusion and inclusion criteria have been defined, they should be described

In table 1, the title should be changed, it contains not only demographic data

In the abstract, in the part concerning the results, you should put numerical values.

Reviewer 3 Report

This manuscript reports an interesting field experiment of the diet-related disparities and childcare food environments for vulnerable children in South Korea. The data they obtained are precious to some extent. There are still some advices provided for the authors to make a revision. 

1 Firstly, the paper questionnaire is not clear. Secondly, the paper questionnaire is provided, but in Korean, so it must be translated it to English to make it understandable to many Readers, as it is an English-language journal, and to make the scientists all over the world be able to replicate this study.

2 Line130-Line 132, the way of the questions list need to improve, delete “·” and make them in one paragraph.

3 Diet-related disparities only appear in the Community Childcare Centers? Does it happen in families with different income? And how could these CCCs represent most of the conditions of diet-related disparities? 

4 The topic of the manuscript is interesting and actual, but the aim of the study is not clearly stated nor in the abstract neither in the manuscript.

5 Data was collected in 2019, while now is 2023, during the four years, the pandemic had an effect on human living, if diet-related disparities were affected by this?

6 In discussion part, 1) 4.1 need adding more references to improve this part’s discuss depth. 2) 4.1 contains 2 paragraphs, but 4.2 contains 7 paragraphs, this structure shows unbalance, it needs more improvement.

Reviewer 4 Report

The topic of this article is of interest, but minor changes are necessary in order to be published.

Results

I suggest to compare the anthropometric parameters as body mass index of the children attending the CCCs before their inclusion in the “second home” and after a determined period of time (1 year, 2 years etc.) in order to obtain a better quantification of the results.  

Discussion

I suggest to compare the results of the actual study to the diet quality of other groups of children from other after-school care spaces from other area.  

Round 2

Reviewer 1 Report

Thanks to the authors for understanding the suggestions. Now the text is more understandable. That's fine with me. I defer the final decision to Editor.